# Complex Gated Recurrent Neural Networks

**Moritz Wolter**
Institute for Computer Science
University of Bonn
wolter@cs.uni-bonn.de

**Angela Yao**
School of Computing
National University of Singapore
yaoa@comp.nus.edu.sg

## Abstract

Complex numbers have long been favoured for digital signal processing, yet complex representations rarely appear in deep learning architectures. RNNs, widely used to process time series and sequence information, could greatly benefit from complex representations. We present a novel complex gated recurrent cell, which is a hybrid cell combining complex-valued and norm-preserving state transitions with a gating mechanism. The resulting RNN exhibits excellent stability and convergence properties and performs competitively on the synthetic memory and adding task, as well as on the real-world tasks of human motion prediction.

## 1 Introduction

Recurrent neural networks (RNNs) are widely used for processing time series and sequential information. The difficulties of training RNNs, especially when trying to learn long-term dependencies, are well-established, as RNNs are prone to vanishing and exploding gradients [2, 12, 31]. Heuristics developed to alleviate some of the optimization instabilities and learning difficulties include gradient clipping [9, 29], gating [4, 12], and using norm-preserving state transition matrices [1, 13, 16, 40].

Gating, as used in gated recurrent units (GRUs) [4] and long short-term memory (LSTM) networks [12], has become common-place in recurrent architectures. Gates facilitate the learning of longer term temporal relationships [12]. Furthermore, in the presence of noise in the input signal, gates can protect the cell state from undesired updates, thereby improving overall stability and convergence.

A matrix $\mathbf{W}$ is norm-preserving if its repeated multiplication with a vector leaves the vector norm unchanged, *i.e.* $\|\mathbf{W}h\|_2 = \|h\|_2$. Norm-preserving state transition matrices are particularly interesting for RNNs because they preserve gradients over time [1], thereby preventing both the vanishing and exploding gradient problem. To be norm-preserving, state transition matrices need to be either orthogonal or unitary[1]. Complex numbers have long been favored for signal processing [11, 24, 27]. A complex signal does not simply double the dimensionality of the signal. Instead, the representational richness of complex signals is rooted in its physical relevance and the mathematical theory of complex analysis. Complex arithmetic, and in particular multiplication, is different from its real counterpart and allows us to construct novel network architectures with several desirable properties. Despite networks being complex-valued, however, it is often necessary to work with real-valued cost functions and/or existing real-valued network components. Mappings from $\mathbb{C} \to \mathbb{R}$ are therefore indispensable. Unfortunately such functions violate the Cauchy-Riemann equations and are not complex-differentiable in the traditional sense. We advocate the use of Wirtinger calculus [39] (also known as CR-calculus [21]), which makes it possible to define complex (partial) derivatives, even when working with non-holomorph or non-analytic functions.

Complex-valued representations have begun receiving some attention in the the deep learning community but they have been applied only to the most basic of architectures [1, 10, 36]. For recurrent networks, complex representations could gain more acceptance if they were shown to be compatible with more commonly used gated architectures and also competitive for real-world data. This is exactly the aim of this work, where we propose a complex-valued gated recurrent network and show how it can easily be implemented with a standard deep learning library such as TensorFlow. Our contributions can be summarized as follows[2]:

- We introduce a novel complex-gated recurrent unit; to the best of our knowledge, we are the first to explore such a structure using complex number representations.

- We compare experimentally the effects of a bounded versus unbounded non-linearity in recurrent networks, finding additional evidence countering the commonly held heuristic that only bounded non-linearities should be applied in RNNs. In our case unbounded non-linearities perform better, but must be coupled with the stabilizing measure of using norm-preserving state transition matrices.

- Our complex gated network is stable and fast to train; it outperforms the state of the art with equal parameters on synthetic tasks and delivers state-of-the-art performance one the real-world application of predicting poses in human motion capture using fewer weights.

## 2 Related work

The current body of literature in deep learning focuses predominantly on real-valued neural networks. Theory for learning with complex-valued data, however, was established long before the breakthroughs of deep learning. This includes the development of complex non-linearities and activation functions [7, 18], the computation of complex gradients and Hessians [37], and complex backpropagation [3, 23].

Complex-valued representations were first used in deep networks to model phase dependencies for more biologically plausible neurons [33] and to augment the memory of LSTMs [5], *i.e.* whereby half of the cell state is interpreted as the imaginary component. In contrast, true complex-valued networks (including this work) have not only complex valued states but also kernels. Recently, complex CNNs have been proposed as an alternative for classifying natural images [10, 36] and inverse mapping of MRI signals [38]. Complex CNNs were found to be competitive or better than state-of-the-art [36] and significantly less prone to over-fitting [10].

For temporal sequences, complex-valued RNNs have also been explored [1, 13, 17, 40], though interest in complex representations stems from improved learning stability. In [1], norm-preserving state transition matrices are used to prevent vanishing and exploding gradients. Since it is difficult to parameterize real-valued orthogonal weights, [1] recommends shifting to the complex domain, resulting in a unitary RNN (uRNN). The weights of the uRNN in [1], for computational efficiency, are constructed as a product of component unitary matrices. As such, they span only a reduced subset of unitary matrices and do not have the expressiveness of the full set. Alternative methods of parameterizing the unitary matrices have been explored [13, 17, 40]. Our proposed complex gated RNN (cgRNN) builds on these works in that we also use unitary state transition matrices. In particular, we adopt the parameterization of [40] in which weights are parameterized by full-dimensional unitary matrices, though any of the other parameterizations [1, 13, 17] can also be substituted.

## 3 Preliminaries

We represent a complex number $z \in \mathbb{C}$ as $z = x + ib$, where $x = \Re(z)$ and $y = \Im(z)$ are the real and imaginary parts respectively. The complex conjugate of $z$ is $\bar{z} = x - iy$. In polar coordinates, $z$ can be expressed as $z = |z|e^{i\theta_z}$, where $|z|$ and $\theta$ are the magnitude and phase respectively and $\theta_z = \text{atan2}(x, y)$. Note that $z_1 \cdot z_2 = |z_1||z_2|e^{i(\theta_1 + \theta_2)}$, $z_1 + z_2 = x_1 + x_2 + i(y_1 + y_2)$ and $s \cdot z = s \cdot re^{i\theta}, s \in \mathbb{R}$. The expression $s \cdot z$ scales $z$'s magnitude, while leaving the phase intact.

## 3.1 Complex Gradients

A complex-valued function $f : \mathbb{C} \to \mathbb{C}$ can be expressed as $f(z) = u(x, y) + iv(x, y)$ where $u(\cdot, \cdot)$ and $v(\cdot, \cdot)$ are two real-valued functions. The complex derivative of $f(z)$, or the $\mathbb{C}$-derivative, is defined if and only if $f$ is *holomorph*. In such a case, the partial derivatives of $u$ and $v$ must not only exist but also satisfy the Cauchy-Riemann equations, where $\partial u / \partial x = \partial v / \partial y$ and $\partial v / \partial x = -\partial u / \partial y$.

Strict holomorphy can be overly stringent for deep learning purposes. In fact, Liouville's theorem [25] states that the only complex function which is both holomorph and bounded is a constant function. This implies that for complex (activation) functions, one must trade off either boundedness or differentiability. One can forgo holomorphy and still leverage the theoretical framework of Wirtinger or CR-calculus [21, 27] to work separately with the $\mathbb{R}$- and $\overline{\mathbb{R}}$- derivatives[3]:

$$\mathbb{R}\text{-derivative} \triangleq \frac{\partial f}{\partial z}|_{\bar{z}=\text{const}} = \frac{1}{2}\left(\frac{\partial f}{\partial x} - i\frac{\partial f}{\partial y}\right), \ \overline{\mathbb{R}}\text{-derivative} \triangleq \frac{\partial f}{\partial \bar{z}}|_{z=\text{const}} = \frac{1}{2}\left(\frac{\partial f}{\partial x} + i\frac{\partial f}{\partial y}\right). \quad (1)$$

Based on these derivatives, one can define the chain rule for a function $g(f(z))$ as follows:

$$\frac{\partial g(f(z))}{\partial z} = \frac{\partial g}{\partial f}\frac{\partial f}{\partial z} + \frac{\partial g}{\partial \bar{f}}\frac{\partial \bar{f}}{\partial z} \qquad \text{where} \qquad \bar{f} = u(x, y) - iv(x, y). \quad (2)$$

Since mappings from $\mathbb{C} \to \mathbb{R}$ can generally be expressed in terms of the complex variable $z$ and its conjugate $\bar{z}$, the Wirtinger-Calculus allows us to formulate and theoretically understand the gradient of real-valued loss functions in an easy yet principled way.

## 3.2 A Split Complex Approach

We work with a split-complex approach, where real-valued non-linear activations are applied separately to the real and imaginary parts of the complex number. This makes it convenient for implementation, since standard deep learning libraries are not designed to work natively with complex representations. Instead, we store complex numbers as two real-valued components. Split-complex activation functions process either the magnitude and phase, or the real and imaginary components with two real-valued nonlinear functions and then recombine the two into a new complex quantity. While some may argue this reduces the utility of having complex representations, we prefer this to fully complex activations. Fully complex non-linearities do exist and may seem favorable [36], since one needs to keep track of only the $\mathbb{R}$ derivatives, but due to Liouville's theorem, we must forgo boundedness and then deal with forward pass instabilities.

# 4 Complex Gated RNNs

## 4.1 Basic Complex RNN Formulation

Without any assumptions on real versus complex representations, we define a basic RNN as follows:

$$\mathbf{z}_t = \mathbf{W}\mathbf{h}_{t-1} + \mathbf{V}\mathbf{x}_t + \mathbf{b} \quad (3)$$
$$\mathbf{h}_t = f_a(\mathbf{z}_t) \quad (4)$$

where $\mathbf{x}_t$ and $\mathbf{h}_t$ represent the input and hidden unit vectors at time $t$. $f_a$ is a point-wise non-linear activation function, and $W$ and $V$ are the hidden and input state transition matrices respectively. In working with complex networks, $\mathbf{x}_t \in \mathbb{C}^{n_x \times 1}$, $\mathbf{h}_t \in \mathbb{C}^{n_h \times 1}$, $\mathbf{W} \in \mathbb{C}^{n_h \times n_h}$, $\mathbf{V} \in \mathbb{C}^{n_h \times n_x}$ and $\mathbf{b} \in \mathbb{C}^{n_h \times 1}$, where $n_x$ and $n_h$ are the dimensionalities of the input and hidden states respectively.

## 4.2 Complex Non-linear Activation Functions

Choosing a non-linear activation function $f_a$ for complex networks can be non-trivial. Though holomorph non-linearities using transcendental functions have also been explored in the literature [27], the presence of singularities makes them difficult to learn in a stable manner. Instead, bounded non-holomorph non-linearities tend to be favoured [11, 27], where bounded real-valued non-linearities are applied on the real and imaginary part separately. This also parallels the convention of using (bounded) tanh non-linearities in real RNNs.

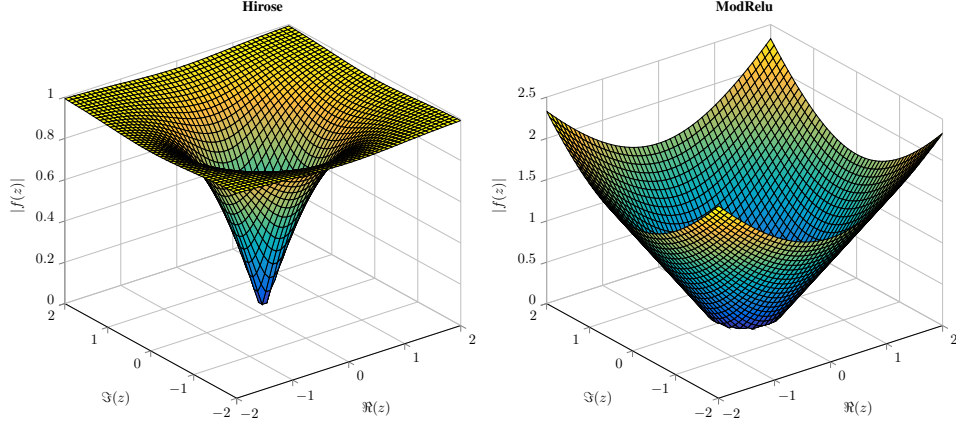

Figure 1: Surface plots of the magnitude of the Hirose ($m^2 = 1$) and modReLU ($b = -0.5$) activations.

A common split is with respect to the magnitude and phase. This non-linearity was popularized by Hirose [11] and scales the magnitude by a factor $m^2$ before passing it through a $\tanh$:

$$f_{\text{Hirose}}(z) = \tanh\left(\frac{|z|}{m^2}\right) e^{-i \cdot \theta_z} = \tanh\left(\frac{|z|}{m^2}\right) \frac{z}{|z|}. \tag{5}$$

In other areas of deep learning, the rectified linear unit (ReLU) is now the go-to non-linearity. In comparison to sigmoid or $\tanh$ activations, they are computationally cheap, expedite convergence [22] and also perform better [30, 26, 42]. However, there is no direct extension into the complex domain, and as such, modified versions have been proposed [10, 38]. The most popular is the modReLU [1] – a variation of the Hirose non-linearity, where the $\tanh$ is replaced with a ReLU and $b$ is an offset:

$$f_{\text{modReLU}}(z) = \text{ReLU}(|z| + b)e^{-i \cdot \theta_z} = \text{ReLU}(|z| + b)\frac{z}{|z|}. \tag{6}$$

### 4.3 $\mathbb{R} \to \mathbb{C}$ input and $\mathbb{C} \to \mathbb{R}$ output mappings

While several time series problems are inherently complex, especially when considering their Fourier representations, the majority of benchmark problems in machine learning are still only defined in the real number domain. However, one can still solve these problems with complex representations, since a real $z$ has simply a zero imaginary component, *i.e.* $\Im(z) = 0$ and $z = x + i \cdot 0$.

To map the complex state $\mathbf{h}$ into a real output $\mathbf{o}_r$, we use a linear combination of the real and imaginary components, similar to [1], with $\mathbf{W}_o$ and $\mathbf{b}_o$ as weights and offset:

$$\mathbf{o}_r = \mathbf{W}_o[\Re(\mathbf{h}) \; \Im(\mathbf{h})] + \mathbf{b}_o. \tag{7}$$

### 4.4 Optimization on the Stiefel Manifold for Norm Preservation

In [1], it was proven that a unitary [4] $\mathbf{W}$ would prevent vanishing and exploding gradients of the cost function $C$ with respect to $h_t$, since the gradient magnitude is bounded. However, this proof hinges on the assumption that the derivative of $f_a$ is also unity. This assumption is valid if the pre-activations are real and one chooses the ReLU as the non-linearity. For complex pre-activations, however, this is no longer a valid assumption. Neither the Hirose non-linearity (Equation 5) nor the modReLU (Equation 6) can guarantee stability (despite the suggestion otherwise in the original proof [1]).

Even though it is not possible to guarantee stability, we strongly advocate using norm-preserving state transition matrices, since they do still have excellent stabilizing effects. This was proven experimentally in [1, 13, 40] and we find similar evidence in our own experiments (see Figure 2). Ensuring that $\mathbf{W}$ remains unitary during the optimization can be challenging, especially since the group of unitary matrices is not closed under addition. As such, it is not possible to learn $\mathbf{W}$ with

standard update-based gradient descent. Alternatively, one can learn $\mathbf{W}$ on the Stiefel manifold [40], with the $k + 1$ update $\mathbf{W}_{k+1}$ given as follows by [34], where $\lambda$ is the learning rate, $\mathbf{I}$ the identity matrix, and $F$ the cost function:

$$\mathbf{W}_{k+1} = (\mathbf{I} + \frac{\lambda}{2}\mathbf{A}_k)^{-1}(\mathbf{I} - \frac{\lambda}{2}\mathbf{A}_k)\mathbf{W}_k \qquad \text{where} \qquad \mathbf{A} = \mathbf{W}\overline{\nabla_{\mathbf{w}}F}^T - \overline{\mathbf{W}}^T\nabla_{\mathbf{w}}F. \quad (8)$$

### 4.5 Complex-Valued Gating Units

In keeping with the spirit that gates determine the amount of a signal to pass, we construct a complex gate as a $\mathbb{C}^{n_h \times n_h} \to \mathbb{R}^{n_h \times 1}$ mapping. Like in real gated RNNs, the gate is applied as an element-wise product, $i.e.\mathbf{g} \odot \mathbf{h} = \mathbf{g} \odot |\mathbf{h}|e^{i\theta_h}$. In our complex case, this type of operation results in an element-wise scaling of the hidden state's magnitude. When the gate is 0, it completely resets a signal, whereas if it is 1, then it ensures that the signal is passed entirely. We introduce our gates into the RNN in a similar fashion as the classic GRU [4]:

$$\widetilde{\mathbf{z}}_t = \mathbf{W}(\mathbf{g}_r \odot \mathbf{h}_{t-1}) + \mathbf{V}\mathbf{x}_t + \mathbf{b}, \quad (9)$$
$$\mathbf{h}_t = \mathbf{g}_z \odot f_a(\widetilde{\mathbf{z}}_t) + (1 - \mathbf{g}_z) \odot \mathbf{h}_{t-1}, \quad (10)$$

where $\mathbf{g}_r$ and $\mathbf{g}_z$ represent reset and update gates respectively and are defined with corresponding subscripts $r$ and $z$ as

$$\mathbf{g}_r = f_g(\mathbf{z}_r), \qquad \text{where} \qquad \mathbf{z}_r = \mathbf{W}_r\mathbf{h} + \mathbf{V}_r\mathbf{x}_t + \mathbf{b}_r, \quad (11)$$
$$\mathbf{g}_z = f_g(\mathbf{z}_z), \qquad \text{where} \qquad \mathbf{z}_z = \mathbf{W}_z\mathbf{h} + \mathbf{V}_z\mathbf{x}_t + \mathbf{b}_z. \quad (12)$$

Above, $f_g$ denotes the gate activation, $\mathbf{W}_r \in \mathbb{C}^{n_h \times n_h}$ and $\mathbf{W}_z \in \mathbb{C}^{n_h \times n_h}$ denote state to state transition matrices, $\mathbf{V}_r \in \mathbb{C}^{n_h \times n_i}$ and $\mathbf{V}_z \in \mathbb{C}^{n_h \times n_i}$ the input to state transition matrices, and $\mathbf{b}_r \in \mathbb{C}^{n_h}$ and $\mathbf{b}_z \in \mathbb{C}^{n_h}$ the biases. $f_g$ is a non-linear gate activation function defined as:

$$f_{\text{mod sigmoid}}(\mathbf{z}) = \sigma(\alpha\Re(\mathbf{z}) + \beta\Im(\mathbf{z})), \qquad \alpha, \beta \in [0, 1]. \quad (13)$$

We call this the modSigmoid and justify the choice experimentally in section 5.3.

As mentioned previously, even with unitary state transition matrices, this type of gating is not mathematically guaranteed to be stable. However, the effects of vanishing gradients are mitigated by the fact that the derivatives are distributed over a sum [12, 4]. Exploding gradients are clipped.

## 5 Experimentation

### 5.1 Tasks & Evaluation Metrics

We test our cgRNN on two benchmark synthetic tasks: the memory problem and the adding problem [12]. These problems are designed especially to challenge RNNs, and require the networks to store information over time scales on the order of hundreds of time steps. The first is the **memory problem**, where the RNN should remember $n$ input symbols over a time period of length $T + 2n$ based on a dictionary set $\{s^1, s^2, ..., s^n, s^b, s^d\}$, where $s^1$ to $s^n$ are symbols to memorize and $s^b$ and $s^i$ are blank and delimiter symbols respectively. The input sequence, of length $T + 2n$, is composed of $n$ symbols drawn randomly with replacement from $\{s^1, ..., s^n\}$, followed by $T - 1$ repetitions of $s^b, s^d$, and another $n$ repetitions of $s^b$. The objective of the RNN, after being presented with the initial $n$ symbols, is to generate an output sequence of length $T + S$, with $T$ repetitions of $s^b$, and upon seeing $s^d$, recall the original $n$ input symbols. A network without memory would output $s^b$ and once presented with $s^d$, randomly predict any of the original $n$ symbols; this results in a categorical cross entropy of $\frac{(n + 1\log(8))}{(T + 2(n + 2))}$. For our experiments, we choose $n = 8$ and $T = 250$.

In the **adding problem**, two sequences of length $T$ are given as input, where the first sequence consists of numbers randomly sampled from $\mathcal{U}[0, 1]^5$, while the second is an indicator sequence of all $0's$ and exactly two $1's$, with the first 1 placed randomly in the first half of the sequence and the second 1 randomly in the second half. The objective of the RNN is to predict the sum of the two entries of the first input sequence once the second 1 is presented in the indicator input sequence. A

naive baseline would predict 1 at every time step, regardless of the input indicator sequence's value; this produces an mean squared error (MSE) of 0.167, *i.e.* the variance of the sum of two independent uniform distributions. For our experiments, we choose $T = 250$.

We apply the cgRNN to the real-world task of **human motion prediction**, *i.e.* predicting future 3D poses of a person given the past motion sequence. This task is of interest to diverse areas of research, including 3D tracking in computer vision [41], motion synthesis for graphics [20] as well as pose and action predictions for assistive robotics [19]. We follow the same experimental setting as [28], working with the full Human 3.6M dataset [14]. For training, we use six of the seven actors and test on actor five. We use the pre-processed data of [15], which converts the motion capture into exponential map representations of each joint. Based on an input sequence of body poses from 50 frames, the future 10 frames are predicted. This is equivalent of predicting 400ms. The error is measured by the euclidean distance in Euler angles with respect to the ground truth poses.

We also test the cgRNN on native complex data drawn from the frequency domain by testing it on the real world task of **music transcription**. Given a music wave form file, the network should determine the notes of each instrument. We use the Music-Net dataset [35], which consists of 330 classical music recordings, of which 327 are used for training and 3 are held out for testing. Each recording, sampled at 11kHz, is divided into segments of 2048 samples with a step size of 512 samples. The transcription problem is defined as a multi-label classification problem, where for each segment, a label vector $y \in 0, 1^{128}$ describing the active keys in the corresponding midi file has to be found. We use the windowed Fourier-transform of each segment as network input, the real and imaginary parts of the Fourier transform, *i.e.* the odd and even components respectively, are used directly as inputs into the cgRNN.

## 5.2    RNN Implementation Details

We work in Tensorflow, using RMS-prop to update standard weights and the multiplicative Stiefel-manifold update as described Equation 8 for all unitary state transition matrices. The unitary state transition matrices are initialized the same as [1] as the product of component unitary matrices. All other weights are initialized using the uniform initialisation method recommended in [8], *i.e.* $U[-l, l]$ with $l = \sqrt{6/(n_{in} + n_{out})}$, where $n_{in}$ and $n_{out}$ are the input and output dimensions of the tensor to be initialised. All biases are intialized as zero, with the exception of the gate biases $\mathbf{b}_r$ and $\mathbf{b}_z$, which are initialized at 4 to ensure fully open gates and linear behaviour at the start of training. All synthetic tasks are run for $2 \cdot 10^4$ iterations with a batch-size of 50 and a constant learning rate of 0.001 for both the RMS-Prop and the Stiefel-Manifold updates.

For the human motion prediction task, we adopt the state-of-the-art implementation of [28], which introduces residual velocity connections into the standard GRU. Our setup shares these modifications; we simply replace their core GRU cell with our cgRNN cell. The learning rate and batch size are kept the same (0.005, 16) though we reduce our state size to 512 to be compatible with [28]'s $1024^6$. For music transcription, we work with a bidirectional cgRNN encoder followed by a simple cgRNN decoder. All cells are set with $n_h = 1024$; the learning rate is set to 0.0001 and batch size to 5.

## 5.3    Impact of Gating and Choice of Gating Functions

We first analyse the impact that gating has on the synthetic tasks by comparing our cgRNN with the gateless uRNN from [1]. Both networks use complex representations and also unitary state transition matrices. As additional baselines, we also compare with TensorFlow's out-of-the-box GRU. We choose the hidden state size $n_h$ of each network to ensure that the resulting number of parameters is approximately equivalent (around $44k$). We find that our cgRNN successfully solves both the memory problem as well as the adding problem. On the memory problem (see Figure 2(a), Table 1), gating does not play a role. Instead, having norm-preserving weight matrices is key to ensure stability during the learning. The GRU, which does not have norm-preserving state matrices, is highly unstable and fails to solve the problem. Our cgRNN achieves very similar performance as the uRNN. This has to do with the fact that we initialize our gate bias term to be fully open, *i.e.* $\mathbf{g}_r = \mathbf{1}$, $\mathbf{g}_z = \mathbf{1}$. Under this setting, the formulation is the same as the uRNN, and the unitary $\mathbf{W}$ dominates the cell's dynamics.

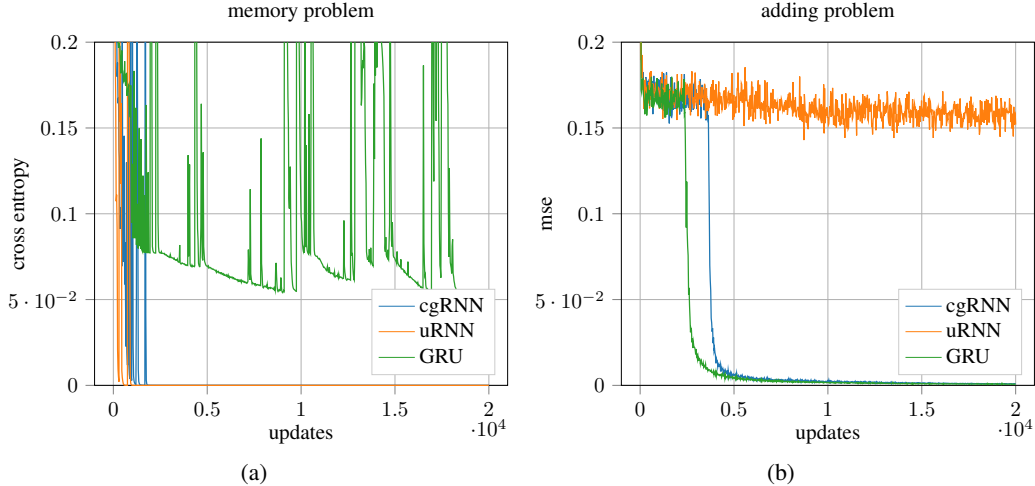

<div align="center">(a)            (b)</div>

Figure 2: Comparison of our cgRNN (blue, $n_h = 80$) with the uRNN [1] (orange, $n_h = 140$) and standard GRU [4] (green, $n_h = 112$) on the memory (a) and adding (b) problem for $T = 250$. The hidden state size $n_h$ for each network are chosen so as to approximately match the number of parameters (approximately 44k parameters total). On the memory problem, having norm-preserving state transition matrices is critical for stable learning, while on the adding problem, having gates is important. Figure best viewed in colour.

For the adding problem, previous works [1, 13, 40] have suggested that gates are beneficial and we confirm this result in Figure 2(b) and Table 1. We speculate that the advantage comes from the gates shielding the network from the irrelevant inputs of the adding problem, hence the success of our cgRNN as well as the GRU, but not the uRNN. Surprisingly, the standard GRU baseline, without any norm-preserving state transition matrices works very well on the adding problem; in fact, it marginally outperforms our cgRNN. However, we believe this result does not speak to the inferiority of complex representations; instead, it is likely that the adding problem, as a synthetic task, is not able to leverage the advantages offered by the representation.

The gating function (Equation 13) was selected experimentally based on a systematic comparison of various functions. The performance of different gate functions are compared statistically in Table 1, where we look at the fraction of converged experiments over 20 runs as well as the mean number of iterations required until convergence. The product as well as the tied and free weighted sum variations of the gating function are designed to resemble the bilinear gating mechanism used in [6]. From our experiments, we find that it is important to scale the real and imaginary components before passing through the sigmoid to leverage the saturation constraint, and that the real and imaginary components should be combined linearly. The exact weighting seems not to be important and the best performing

<div align="center">Table 1: Comparison of gating functions on the adding and memory problems.</div>

| | | gating function | memory problem frac.conv. | memory problem avg.iters. | adding problem frac.conv. | adding problem avg.iters. |
|---|---|---|---|---|---|---|
| uRNN [40] | | no gate | 1.0 | 2235 | 0.0 | - |
| cgRNN | product | $\sigma(\Re(\mathbf{z}))\sigma(\Im(\mathbf{z}))$ | 0.10 | 4625 | 1.0 | 4245 |
| | tied 1 | $\alpha\sigma(\Re(\mathbf{z})) + (1-\alpha)\sigma(\Im(\mathbf{z}))$ | 0.55 | 4186 | 1.0 | 5458 |
| | tied 2 | $\sigma(\alpha\Re(\mathbf{z}) + (1-\alpha)\Im(\mathbf{z}))$ | 0.80 | 3800 | 1.0 | 5070 |
| | free | $\sigma(\alpha\Re(\mathbf{z}) + \beta\Im(\mathbf{z}))$ | 0.75 | 2850 | 1.0 | 5235 |
| free real | | $\sigma(\alpha\mathbf{z}_1 + \beta\mathbf{z}_2), (\mathbf{z}_1, \mathbf{z}_2) \in \mathbb{R}$ | 0.0 | - | 1.0 | 5313 |

The different gates are evaluated over 20 runs by looking at the fraction of convergence (frac.conv.) and average number of iterations required for convergence (avg.iters.) if convergent. A run is considered convergent if the loss falls below $5 \cdot 10^{-7}$ for the memory problem and 0.01 for the adding problem. We find that gating has no impact for the memory problem, *i.e.* the gateless uRNN [40] always converges, but is necessary for the adding problem. All experiments use weight normalized recurrent weights, a cell size of $n_h = 80$, and have networks with approximately 44k parameters; to keep approximately the same number of parameters, we set $n_h = 140$ for the uRNN and two independent gates each with $n_h = 90$ for the real free real case.

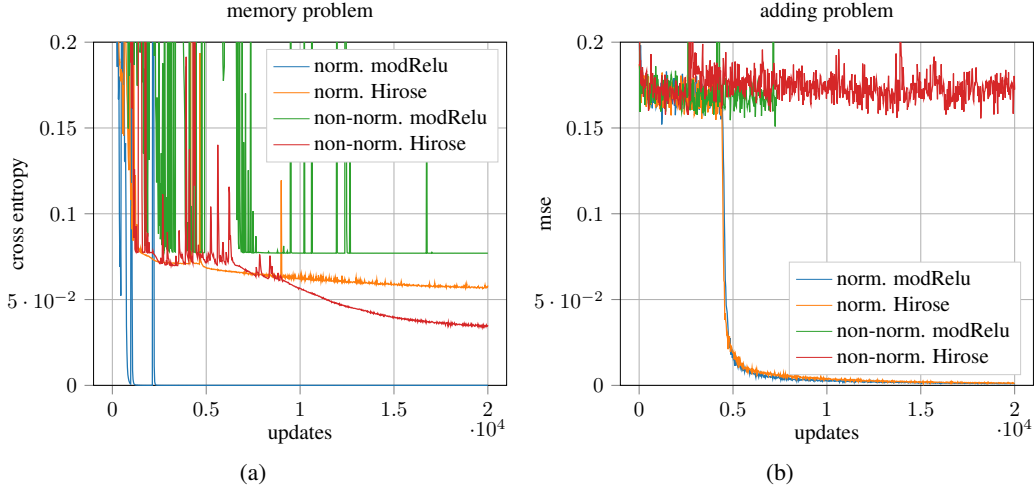

Figure 3: Comparison of non-linearities and norm preserving state transition matrices on the cgRNNs for the memory (a) and adding (b) problems for T=250. The unbounded modReLU (see Equation 6) performs best for both problems, but only if the state transition matrices are kept unitary. Without unitary state-transition matrices, the bounded Hirose non-linearity (see Equation 5) performs better. We use $n_h = 80$ for all experiments.

variants are the *tied 2* and the *free*; to preserve generality, we advocate the use of the *free* variant. We note that over 20 runs, our cgRNN converged only on 15-16 runs; adding the gates introduces instabilities, however, we find the ability to solve the adding problem a reasonable trade-off.

Finally, we compare the cgRNN to a *free real* variant (see last row of Table 1), which is the most similar architecture in $\mathbb{R}$, *i.e.*normalized hidden transition matrices, same gate formulation, and two independently real-valued versions of Equations 11 and 12. This real variant has similar performance on the adding problem (for which having gates is critical), but cannot solve the memory problem. This is likely due to the set of real orthogonal matrices being too restrictive, making the problem more difficult in the real domain than the complex.

### 5.4  Non-Linearity Choice and Norm Preservation

We compare the bounded Hirose $\tanh$ non-linearity versus the unbounded modReLU (see Section 4.2) in our cgRNN in Figure 3 and discover a strong interaction effect from the norm-preservation. First, we find that optimizing on the Stiefel manifold to preserve norms for the state transition matrices significantly improves learning, regardless of the non-linearity. In both the memory and the adding problem, keeping the state transition matrices unitary ensures faster and smoother convergence of the learning curve.

Without unitary state transition matrices, the bounded $\tanh$ non-linearity, *i.e.*the conventional choice is better than the unbounded modReLU. However, with unitary state transition matrices, the modReLU pulls ahead. We speculate that the modReLU, like the ReLU in the real setting, is a better choice of non-linearity. The advantages afforded upon it by being unbounded, however, also makes it more sensitive to stability, which is why these advantages are present only when the state-transition matrices are kept unitary. Similar effects were observed in real RNNs in [32], in which batch normalization was required in order to learn a standard RNN with the ReLU non-linearity.

### 5.5  Real World Tasks: Human Motion Prediction & Music Transcription

We compare our cgRNN to the state of the art GRU proposed by [28] on the task of human motion prediction, showing the results in Table 2. Our cgRNN delivers state-of-the-art performance, while reducing the number of network parameters by almost 50%. However this reduction comes at the cost of having to compute the matrix inverse in Equation 8.

On the music transcription task, we are able to accurately transcribe the input signals with an accuracy of 53%. While this falls short of the complex convolutional state-of-the-art 72.9% of [36], their

complex convolution-based network is fundamentally different from our approach. We conclude that our cgRNN is able to extract meaningful information from complex valued input data and will look into integrating complex convolutions into our RNN as future work.

Table 2: Comparison of our cgRNN with the GRU [28] on human motion prediction.

| Action | cgRNN | | | | GRU[28] | | | |
|---|---|---|---|---|---|---|---|---|
| | 80ms | 160 ms | 320ms | 400ms | 80ms | 160ms | 320ms | 400ms |
| walking | 0.29 | 0.48 | 0.74 | 0.84 | **0.27** | **0.47** | **0.67** | **0.73** |
| eating | 0.23 | **0.38** | 0.66 | 0.82 | 0.23 | 0.39 | **0.62** | **0.77** |
| smoking | **0.31** | **0.58** | **1.01** | **1.1** | 0.32 | 0.6 | 1.02 | 1.13 |
| discussion | 0.33 | 0.72 | **1.02** | **1.08** | **0.31** | **0.7** | 1.05 | 1.12 |
| directions | 0.41 | **0.65** | **0.83** | **0.93** | 0.41 | 0.65 | 0.83 | 0.96 |
| greeting | 0.53 | 0.87 | **1.26** | **1.43** | 0.52 | **0.86** | 1.30 | 1.47 |
| phoning | **0.58** | 1.09 | 1.57 | 1.72 | 0.59 | **1.07** | **1.50** | **1.67** |
| posing | **0.37** | **0.72** | **1.38** | **1.65** | 0.64 | 1.16 | 1.82 | 2.1 |
| purchases | 0.61 | 0.86 | **1.21** | 1.31 | **0.6** | **0.82** | 1.13 | **1.21** |
| sitting | 0.46 | 0.75 | 1.22 | **1.44** | **0.44** | **0.73** | **1.21** | 1.45 |
| sitting down | 0.55 | 1.02 | 1.54 | 1.73 | **0.48** | **0.89** | **1.36** | **1.57** |
| taking photo | **0.29** | **0.59** | **0.92** | **1.07** | 0.29 | 0.59 | 0.95 | 1.1 |
| waiting | 0.35 | 0.68 | 1.16 | **1.36** | **0.33** | **0.65** | **1.14** | 1.37 |
| walking dog | 0.57 | 1.09 | 1.45 | 1.55 | **0.54** | **0.94** | **1.32** | **1.49** |
| walking together | **0.27** | **0.53** | **0.77** | **0.86** | 0.28 | 0.56 | 0.8 | 0.88 |
| average | **0.41** | **0.73** | **1.12** | **1.26** | 0.42 | 0.74 | **1.12** | 1.27 |

Our cgRNN ($n_h = 512$, 1.8M params) predicts human motions which are either comparable or slightly better than the real-valued GRU [28] ($n_h = 1024$, 3.4M params) despite having only approximately half the parameters.

## 6 Conclusion

In this paper, we have proposed a novel complex gated recurrent unit which we use together with unitary state transition matrices to form a stable and fast to train recurrent neural network. To enforce unitarity, we optimize the state transition matrices on the Stiefel manifold, which we show to work well with the modReLU. Our complex gated RNN achieves state-of-the-art performance on the adding problem while remaining competitive on the memory problem. We further demonstrate the applicability of our network on real-world tasks. In particular, for human motion prediction we achieve state-of-the-art performance while significantly reducing the number of weights. The experimental success of the cgRNN leads us to believe that complex representations have significant potential and advocate for their use not only in recurrent networks but in deep learning as a whole.

**Acknowledgements:** Research was supported by the DFG project YA 447/2-1 (DFG Research Unit FOR 2535 Anticipating Human Behavior). We also gratefully acknowledge NVIDIA's donation of a Titan X Pascal GPU.

## Footnotes

[1]Unitary matrices are the complex analogue of orthogonal matrices, *i.e.* a complex matrix $\mathbf{W}$ is unitary if $\mathbf{W}\overline{\mathbf{W}}^T = \overline{\mathbf{W}}^T \mathbf{W} = I$, where $\overline{\mathbf{W}}^T$ is its conjugate transpose and $\mathbf{I}$ is the identity matrix.

[2]Source code available at `https://github.com/v0lta/Complex-gated-recurrent-neural-networks`

[3]For holomorph functions the $\overline{\mathbb{R}}$-derivative is zero and the $\mathbb{C}$- derivative is equal to the $\mathbb{R}$-derivative.

[4]Since $\mathbb{R} \subseteq \mathbb{C}$, we use the term unitary to refer to both real orthogonal and complex unitary matrices and make a distinction for clarity purposes only where necessary.

[5]Note that this is a variant of [12]'s original adding problem, which draws numbers from $\mathcal{U}[-1, 1]$ and used three indicators $\{-1, 0, 1\}$. Our variant is consistent with state-of-the-art [1, 13, 40]

[6]This reduction is larger than necessary – parameter-wise, the equivalent state size is $\sqrt{\frac{1024^2}{2}} \approx 724$

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
