[Reviews · NeurIPS 2018]

Reviewer 1



Summary of approach and contributions: The authors resurrect the pioneering work of Hirose on complex valued neural networks in order to provide a new RNN based on a complex valued activation/transition function and a complex argument gating mechanism. In order to obtain a differentiable function that is not constant and yet bounded, the authors step away from holomorphic functions and employ CR calculus. The authors show experimental improvements on two synthetic tasks and one actual data set. Strengths of the paper: o) Moving away from strict holomorphy and using CR calculus to apply complex valued networks to RNNs is interesting as a novel technique. I think that the authors should spend more time explaining how phases can be easily encoded in the complex domain and therefore why such complex representations can be advantageous for sequential learning. o) The paper is cleanly written and the architectural changes simple enough that they can be written in a few lines of a standard deep learning package such as tensorflow. o) The study of Hirose vs modReLU and the impact of learning transition matrices on the Stiefel Manifold is interesting as it gives additional evidence helping compare two radically different approaches to maintaining stability in a RNN. Such a study fits nicely within recent architectural developments in the field of RNNs. Weaknesses of the paper: o) The authors rely on CR calculus to differentiate the complex functions they employ and, more importantly, design an architecture where key mechanisms (equation (7) and (13)) could have a similar construction with real numbers and additional numbers of dimensions. For instance (13) is analogous to a bilinear gating mechanism which has been employed studied in https://arxiv.org/pdf/1705.03122.pdf for sequential learning. I believe that in order to isolate the effect of learning complex valued values it would be necessary to compare the performance of the proposed architecture with a real valued competitor network that would employ the same family of non-linearities, in particular bi-linear gates as in (13). o) The authors provide very few insights as to why a complex valued representation could be better for sequential learning. I believe that even simple elements of signal processing could expose in a paragraph theoretical elements that would help consider the source of experimental improvement as coming from a complex valued representation. o) The experiments themselves are not most convincing. It seems surprising that no neural translation or neural language understanding benchmark is featured in the paper. The only real-work task should feature the size of the networks involved in numbers of parameters and MB as well as a comparison of computational complexity. Without such elements, it is not certain that better results could be achieved by a larger real valued network. Overall the paper appears original and well written but could really benefit from more substantial theoretical and experimental insights designed to isolate the role of complex values in the network from non-linearity design changes. The author's feedback has better convinced me that complex values play a fundamental positive role in the architecture. I indeed am familiar with the ease of use of complex representations and their expressiveness in the field of time series and the addition of sequential tasks will help that point clearer. In that respect, I appreciate that the authors will add new experimental results to make their experimental claim more robust with TIMIT and WMT data. I revised my rating accordingly.

Reviewer 2



The paper presents the building blocks for a complex valued recurrent neural network with a very well written background and motivation. The background on complex arithmetic and calculus was concise making it readable for people from different backgrounds. I have the following concerns: 1) How does this compare (architecturally) to associative LSTMs (Danihelka et al) without their associative retrieval mechanism? The first claim is that this is the first gated complex RNN, making this distinction with associative LSTMs is important in justifying that claim. 2) “finding evidence countering the commonly held heuristic that only bounded non-linearities should be applied in RNNs.” - Listing this as part of your contributions seems misleading since unbounded non-linearities have used in RNNs in the past. 3) Are gains coming simply from better weight sharing (leading to deeper/wider architectures with the same parameter budget) as in the computer vision experiments in Trabelsi et al 2017, norm preservation or complex parameterization 4) Trabelsi et al and Parcollet et al (this is work appears to have been published after this submission so comparisons are obviously not expected) demonstrate the effectiveness of complex/quaternion models in speech applications, I think exploring real world applications that would directly benefit from a complex parameterization would be interesting. Along the same lines, I think it would also be interesting to show that such a model can learn much faster or with far fewer parameters in settings where learning the phase of a signal is a necessary inductive bias such as in model sin waves. Overall, I like this paper for the well written motivation and concise explanation and background on complex valued calculus. However, my primary concerns are with the experiments, the copy and addition tasks are good sanity checks for the promise of norm preservation, but I think this work needs better real world task evaluations especially with speech datasets.

Reviewer 3



In this paper, the authors propose a recurrent neural network architecture which is based on the GRU but uses complex valued hidden states (but real-valued gates) and unitary transition matrices. The model is compared to relevant baselines on two toy tasks and a human motion prediction problem. The work is mostly well presented and easy to understand, and the design choices well motivated. Some improvements could include: dropping some of the repetitions (e.g. about the restrictions of holomorphic functions), motivating Stiefel manifold optimization over e.g. projected gradient descent, or better motivating the choice of having real-valued gates. The main exception to the above comment is the experimental section, which leaves important questions open. The motion dataset needs to be better presented (in more details). Baseline numbers taken from previous publications would also be welcome. As it is, it looks like this paper's implementation of the GRU baseline does significantly worse than the one presented in Martinez et al. for smaller seeds (80 and 160). Why is that? Finally, while I would not consider the following paper a required citation as it is relatively recent, the authors might find it relevant: QuaterNet: A Quaternion-based Recurrent Model for Human Motion, Pavllo et al., 2018 Some typos: l. 1 -- Complex numbers l. 21 -- Norm-preserving l. 80 -- z = x + iy l. 110 -- forward-pass instabilities l. 184 -- s^b and s^d